# Computational Design of a Chimeric Vaccine against *Plesiomonas shigelloides* Using Pan-Genome and Reverse Vaccinology

**DOI:** 10.3390/vaccines10111886

**Published:** 2022-11-08

**Authors:** Mahnoor Mushtaq, Saifullah Khan, Muhammad Hassan, Alhanouf I. Al-Harbi, Alaa R. Hameed, Khadeeja Khan, Saba Ismail, Muhammad Irfan, Sajjad Ahmad

**Affiliations:** 1Department of Health and Biological Sciences, Abasyn University, Peshawar 25000, Pakistan; 2Institute of Biotechnology and Microbiology, Bacha Khan University, Charsadda 24461, Pakistan; 3Department of Pharmacy, Bacha Khan University, Charsadda 24461, Pakistan; 4Department of Medical Laboratory, College of Applied Medical Sciences, Taibah University, Yanbu 46411, Saudi Arabia; 5Department of Medical Laboratory Techniques, School of Life Sciences, Dijlah University College, Baghdad 59058, Iraq; 6Lahore General Hospital, Lahore 54000, Pakistan; 7Department of Biological Sciences, National University of Medical Sciences, Rawalpindi 46000, Pakistan; 8Department of Oral Biology, College of Dentistry, University of Florida, Gainesville, FL 32611, USA

**Keywords:** multi-epitopes, *Plesiomonas shigelloides*, immunoinformatics, molecular docking

## Abstract

The swift emergence of antibiotic resistance (AR) in bacterial pathogens to make themselves adaptable to changing environments has become an alarming health issue. To prevent AR infection, many ways can be accomplished such as by decreasing the misuse of antibiotics in human and animal medicine. Among these AR bacterial species, *Plesiomonas shigelloides* is one of the etiological agents of intestinal infection in humans. It is a gram-negative rod-shaped bacterium that is highly resistant to several classes of antibiotics, and no licensed vaccine against the aforementioned pathogen is available. Hence, substantial efforts are required to screen protective antigens from the pathogen whole genome that can be subjected easily to experimental evaluations. Here, we employed a reverse vaccinology (RV) approach to design a multi-antigenic epitopes based vaccine against *P. shigelloides*. The complete genomes of *P. shigelloides* were retrieved from the National Center for Biotechnological Information (NCBI) that on average consist of 5226 proteins. The complete proteomes were subjected to different subtractive proteomics filters, and in the results of that analysis, out of total proteins, 2399 were revealed as non-redundant and 2827 as redundant proteins. The non-redundant proteins were further checked for subcellular localization analysis, in which three were localized in the extracellular matrix, eight were outer membrane, and 13 were found in the periplasmic membrane. All surface localized proteins were found to be virulent. Out of a total of 24 virulent proteins, three proteins (flagellar hook protein (FlgE), hypothetical protein, and TonB-dependent hemoglobin/transferrin/lactoferrin family receptor protein) were considered as potential vaccine targets and subjected to epitopes prediction. The predicted epitopes were further examined for antigenicity, toxicity, and solubility. A total of 10 epitopes were selected (GFKESRAEF, VQVPTEAGQ, KINENGVVV, ENKALSQET, QGYASANDE, RLNPTDSRW, TLDYRLNPT, RVTKKQSDK, GEREGKNRP, RDKKTNQPL). The selected epitopes were linked with each other via specific GPGPG linkers in order to design a multi-epitopes vaccine construct, and linked with cholera toxin B subunit adjuvant to make the designed vaccine construct more efficient in terms of antigenicity. The 3D structure of the vaccine construct was modeled ab initio as no appropriate template was available. Furthermore, molecular docking was carried out to check the interaction affinity of the designed vaccine with major histocompatibility complex (MHC-)I (PDB ID: 1L1Y), MHC-II (1KG0), and toll-like receptor 4 ((TLR-4) (PDB: 4G8A). Molecular dynamic simulation was applied to evaluate the dynamic behavior of vaccine-receptor complexes. Lastly, the binding free energies of the vaccine with receptors were estimated by using MMPB/GBSA methods. All of the aforementioned analyses concluded that the designed vaccine molecule as a good candidate to be used in experimental studies to disclose its immune protective efficacy in animal models.

## 1. Introduction

The elevated use of antibiotics in treating human and animal infections as well as in agriculture prompts bacteria to evolve antibiotic resistance, which has contributed to significant economic losses and elevated mortality and morbidity [1]. AR is a continuously emerging process in bacterial species to adapt themselves to the environment [2]. Novel therapeutic technologies against antibiotic resistant bacteria are needed to control the global alarming health issues [3]. Therefore, efforts are required to control these antibiotic resistant bacteria globally. For combating bacterial diseases, vaccines and antibodies can be appropriate options [4,5]. At present, no licensed vaccines are present for several bacterial species involved in hospital acquired infections [6]. However, the development of vaccines is costly, time-consuming, and the chances of screening potential antigenic epitopes are low [7]. Subunit vaccines that target cell fragments contain virulence factors [8], as exemplified by the *meningococcal* outer membrane vesicle (OMV) vaccine, porin/PorA, and the pertussis vaccine. With advancements in genomics, many significant advancements in vaccinology took place. Along with this, shotgun sequencing and bioinformatics tool/servers have aided in predicting antigens localized on pathogens’ surfaces for successful vaccine development [9]. Reverse Vaccinology (RV) has been introduced to target specific localized proteins or antigens using genomic information [10]. RV has been effectively used for the development of a multicomponent *meningococcal* serogroup B vaccine (4CMenB) [11]. Compared to the classical RV, pan-genomic based RV (PGRV) is effective because it screens highly conserved vaccine targets in diverse pathogen strains [12].

Herein, we employed two approaches; subtractive proteomics (SP) and RV to present a best vaccine model against *Plesiomonas shigelloides* with the objective of predicting probable antigenic epitopes from the core genome of *P. shigelloides* and to design a multi-epitope vaccine (MEV) against the pathogen. SP in genomics and proteomics is a technique to search proteomes of bacterial pathogens in quest of vaccine candidates [13,14]. The SP and RV methods can be integrated to identify antigenic epitopes for the design of a chimeric vaccine.

*P. shigelloides* belongs to the family of *Enterobacteriaceae* and mainly causes diarrhea and gastrointestinal infections [15]. Morphologically and chemically, it is a rod-shaped oxidase-positive and has been isolated from freshwater, freshwater fish, shellfish, cattle, goats, swine, cats, dogs, monkeys, vultures, snakes, toads, and humans. About 71% of individuals infected have symptoms of acute illness and abdominal pain. The pathogen transmits from seafood or untreated water and affects 29% of people [16]. These bacteria cause many other illnesses which include sepsis, CNS abnormalities, and vision problems. The pathogen is resistant to chloramphenicol, doxycycline, gentamicin, tetracycline, oxytetracycline, sulfamethoxazole-trimethoprim, novobiocin, florfenicol, streptomycin, azithromycin, and spectinomycin [17]. Several works have been undertaken on vaccine development for the pathogen; however, no commercial vaccine is available [18,19,20]. Among the previous works, carbohydrate antigens are reported as having unique structures for developing diagnostic and vaccine strategies [21]. The product of this work is a designed peptide vaccine for researchers to investigate its immune protection ability in vivo. The findings of the study will increase the vaccine antigens library against *P. shigelloides* as well as fast-tracking the vaccine development process. Also, as the vaccine design is based on proteins that form the core genome of the pathogen, the vaccine is likely to provide cross-protection against all sequenced strains of the bacteria.

## 2. Research Methodology

The flow chart for designing a chimeric vaccine for *P. shigelloides* is schematically demonstrated in Figure 1.

### 2.1. Pre-Screening Phase

#### 2.1.1. Complete Retrieval of *P. shigelloides* Genome

A total of two proteomes of *P. shigelloides* strain were retrieved from the National Center for Biotechnology Information (NCBI) and subjected to SP. The proteins which are present in the pathogen core proteome, host non-similar, and are essential for survival were selected [22]. In this phase, literature reported vaccine properties were considered while selecting protein candidates for vaccine designing [23].

#### 2.1.2. Screening Phase

The initial step in the pre-screening phase was to screen proteins; that show sequence conservation among complete sequenced genomes of the pathogen [24], host non-homologous to avoid autoimmune responses [22], and a check was performed to check those proteins that are essential for growth, survival, localization at the surface, membrane or excreted proteins that are efficiently recognized by the immune system and to generate a specific and fast response [25].

#### 2.1.3. Bacterial Pan-Genome Analysis

The genome was processed using the bacterial pan-genome analysis (BPGA) tool. The core proteins were the primary targets for vaccine development. In the process, redundant proteins, resulting in the redundancy of biochemical function, were not selected [26]. Therefore, only non-redundant proteins were selected during immunoinformatics due to their important cellular functionality [27].

#### 2.1.4. Cd-Hit Analysis (Cluster Data at High Identity with Tolerance)

Pathogen non-redundant core proteins were identified by using the CD-HIT server. Proteins showing 60% sequence homology were removed [28]. Using BLASTp, host non-homologous proteins were identified by employing criteria of sequence identity <30% and bit score of >100 [29]. Sequence identity describes the occurrence of the same nucleotide/amino acid at the same position in aligned sequences. On the other hand, the bit score gives statistical significance to the compared sequences.

#### 2.1.5. Subcellular Localization

The resulting core and non-redundant proteome were then analyzed for extracellular, periplasmic, and outer membrane region proteins (OMPs), respectively [30]. For achieving pathogen exo-proteome and secretome, subcellular localization analysis was applied. The outer membrane, inner membrane, extracellular, cytoplasmic, and unknown proteins were predicted by PSORTb v 3.0 [31].

#### 2.1.6. Vaccine Candidate’s Prioritization Phase

The secretome and exoproteome of the bacteria were further filtered to get virulence factor through BLASTp against the virulence factor database (VFDB) [22]. Proteins showing a minimum of 100 bit score and 30% percent sequence identity were considered [32]. The pathogenic proteins were subjected to physicochemical parameters characterization using the ProtParam web server [33]. Different properties such as molecular weight, aliphatic index, instability index, and GRAVY were determined via ProtParam. The number of transmembrane helices was assessed with the help of TMHMM [34] and HMMTOP [35] with the threshold set to less than 2. After selection, the proteins were aligned with the probiotics proteome to reduce the risk of their inhibition [13]. An online web server, BLASTp was used to assess homology against human (taxid: 9606) and three normal flora strains: *Lactobacillus rhamnosus* (taxid: 47715), *L. casei* (taxid: 1582), and *L. johnsonii* (taxid: 33959). Only proteins that showed a bit score >100, and E-value threshold value <0.005 were selected.

#### 2.1.7. Antigenicity, Allergenicity, and Adhesion Probability Prediction

Using Vaxijen 2.0, the antigenicity of proteins was analyzed [36], and preferred only those proteins having an antigenicity greater than 0.4. Similarly, allergenicity was determined with the help of Allertop 2.0 [36], and adhesion probability was achieved using the online web server Vaxign 2.0 [36]

#### 2.1.8. Immune Cell Epitopes Prediction

The selected proteins were subjected to the epitopes prediction phase and analyzed via the Immune Epitope Database (IEDB) [37]. These epitopes help to stimulate/boost the host immune system. The B-cell epitopes were predicted Bipred 2.0. T-cell epitopes prediction involves the evaluation of B-cell epitopes for their effective binding with molecules of both classes of major histocompatibility complexes [30].

#### 2.1.9. MHcPred Analysis

Through the MHcPred server, a DRB*0101 binding analysis was performed and only those epitopes whose IC_50_ values are less than 100 nM for the DRB*0101 gene were subjected to the next steps. The pathogenicity of epitopes was determined by Virulentpred [38], and antigenicity was assured with the help of Vaxijen 2.0. Afterward, non-allergen sequences were extracted by an online allergen prediction tool, Allertop 2.0 [36].

### 2.2. Multi-Epitopes Peptide Construct

The GPGPG linkers were used to join the predicted B-cell derived T-cell epitopes. Finally, the designed epitopes peptide was joined at the N-terminal of the B subunit of cholera toxin adjuvant using an EAAAK linker [39]. The final construct was checked for physicochemical properties via the ProtParam tool. The 3D structure of the multi-epitopes vaccine construct was modeled using 3Dpro of SCRATCH protein predictor [29]. Loop structures were constructed via Galaxy Loop and were then refined via Galaxy Refine [40] of Galaxy Web. 

#### Disulfide Engineering and Codon Optimization

For achieving a stable vaccine, a disulfide engineering using Design 2.0 was performed and then reverse translated to DNA sequence through JCat server [41]. Lastly, the vaccine construct was cloned into pET-28a (+) vector via the Snap Gene tool (https://www.snapgene.com/ (accessed on 1 June 2022).

### 2.3. Molecular Docking

Molecular docking was carried out for the vaccine with different human immune receptors [42]. For prediction of vaccine interactions with immune receptors, blind docking was performed via the PATCHDOCK server [43]. The docked complexes were accomplished with Fast Interaction Refinement in Molecular Docking (FireDock) [44]. The lowest global energy complex was ranked top and subjected to conformation docked complex was visualized using UCSF Chimera 1.13.1. [45].

### 2.4. Molecular Dynamics Simulation (MDS) Analysis 

Additionally, the designed vaccine was revealed for its dynamics in 200-ns of computer simulations, including system making, pre-processing, and production steps using Assistant model construction with Energy Refinement (AMBER) 20 [46]. Using an antechamber program, the parameters of a receptor with vaccine construct were generated [47]. With the help of the Leap program, the TIP3P solvation box was used for the submersion of complexes [48]. The SHAKE algorithm was used to constrain hydrogen bonds. The simulation trajectories were examined for further structural analysis using the CPPTRAJ module of AMBER and Visual Molecular Dynamics (VMD, USA) tool version 1.9.3.

#### 2.4.1. Binding Free Energies Estimation 

Through the MMPBSA.py module, the MMPBSA binding free energy of vaccine-TLR4 was estimated using the AMBER 20 package [49]. Overall, 100 frames were selected from simulation trajectories and subjected to MM/GBSA equation.

#### 2.4.2. Vaccine Immune Simulation

Through a computer immune simulation server, C-Immune Server, host immune reactions (immunogenicity) against the vaccine construct were characterized [50]. The position-specific score matrix (PSSM) and various other machine learning techniques to predict and study epitope and immune interactions were used.

## 3. Results

### 3.1. Retrieval of P. shigelloides Proteomics, Pan-Proteomics and Redundency Check

In the current study, a total two of completely sequenced genomes of *P. shigelloides* were retrieved from NCBI. The MS-17-188 is a multi-drug resistant strain recovered from catfish in 2017. This strain is responsible for gastrointestinal tract infections and is resistant to chloramphenicol, doxycycline, gentamicin, tetracycline, oxytetracycline, sulfamethoxazole-trimethoprim, novobiocin, florfenicol, streptomycin, azithromycin, and spectinomycin. The NCTC10360 is isolated from patients with diarrhea and shows resistance to many antibiotics. All the retrieved data were in fasta format and the proteome data of the strains were subjected to further study. The size and GC content of each genome is tabulated in Table 1.

The two proteomes were processed in the BPGA tool and the core sequences of the bacteria were extracted. The core proteome consists of dispensable proteins and strain-specific proteins. The proteomic data of strains consist of 5226 total proteins. The proteins contain pathogen core proteome, host non-similarity, and essential proteins. A CD-HIT analysis revealed 2399 non-redundant proteins and 2827 redundant proteins [28]. Non-redundant proteins were selected during immunoinformatics due to their important functionality and were subjected to subcellular localization and virulent analysis.

### 3.2. Subcellular Localization

For the prediction of ex-proteome and secretome, a subcellular localization strategy was applied to get surface proteins that have the qualities of invasion, adherence, and proliferation. The PSORTb tool was used to check the localization of proteins [51]. The majority of proteins were cytoplasmic, three were located in the extracellular (EP), eight were located in the outer membrane (OMP), 13 proteins were found in periplasmic (PP), and some proteins were predicted as unknown. The number of surface localized proteins is listed in Figure 2.

### 3.3. Virulence Proteins Analysis and Transmembrane Helices Analysis

The selected proteins were further analyzed to achieve virulence factors and other targets using the virulence factor database (VFDB) [22]. As virulent proteins are involved in the disease pathway, they can also generate immune responses and thus are considered good targets for vaccine design [52]. In this study, only 24 proteins were found to be virulent proteins as represented in Figure 2. TMHMM and HMMTOP were carried out to check the number of transmembrane helices within proteins [35]. Only proteins harboring zero or one transmembrane helix were selected, as they allow easy purification of the proteins in experimental analysis. Only one sequence was removed due to not fulfilling the transmembrane helices criteria, as shown in Figure 2.

### 3.4. Physiochemical Properties of Proteins

The selected proteins were subjected to physicochemical properties [33]. Different physicochemical parameters were determined, such as molecular weight, aliphatic index, GRAVY, amino acid composition, and the instability index. Those proteins were selected showing a molecular weight <110 kDa and an instability index <40, as such proteins can be easily handled during follow-up experimental work. The low molecular weight proteins are easy to purify from the cell, while stable proteins resist degradation. In this study, one protein was discarded based on the above physiochemical criteria, and all the physicochemical properties are described in Appendix A.

### 3.5. Human and Normal Flora Homology, Antigenicity, Allergenicity, and Adhesion Probability Analysis

Probiotic bacteria are those organisms that produce different enzymes to stop the growth and survival of harmful microbes. They produce different vitamins such as biotin, vitamin K2 and are also a key factor for the secretion of antibodies and regulatory cells that stimulate the host immune response. The proteins were aligned with the reference human proteins, and we selected those proteins that showed non-homology with the human genome to avoid an autoimmune response. In this study, only three protein sequences have shown homology with human proteins and 10 protein sequences have shown similarity with the normal flora of the host. The three proteins were found to be antigenic, non-allergenic and showed an adhesion probability value greater than 0.6, indicating that they are good vaccine candidates. An antigenic value greater than 0.4 refers to the strong ability of the proteins to induce immune responses compared to those with a value less than 0.4. Similarly, a protein can be classified as an adhesive if its value is higher than 0.5 [26]. Among the above-filtered proteins criteria, only 03 proteins (flagellar hook protein FlgE, hypothetical protein, and hemoglobin/transferrin/lactoferrin family receptor) were selected for further study, as shown in Table 2.

### 3.6. Immune Epitopes Prediction

The proteins were first utilized to predict B-cell epitopes in which those B-cell epitopes were selected showing scores greater than 0.8 [37]. Only those epitopes were further subjected to the B-cell derived T-cell epitope which has a common binding with MHC-I and MHC-II. The MHC -I and MHC-II alleles subset used are listed in Appendix A [37]. Only those epitopes having a low percentile rank were selected. Through the MHcPred server, a DRB*0101 binding analysis was performed to select those epitopes showing IC_50_ values <100 nM for the DRB*0101 gene. DRB*0101 belongs to human leukocyte antigen II and is present in most human populations. After this, the epitopes were analyzed for allergenicity, solubility, and toxicity analysis. All of the predicted B and T-cell epitopes with percentile rank IC_50_ predicted scores are mentioned in Appendix A.

### 3.7. Antigenicity, Allergenicity, Solubility, and Toxicity Analysis of Predicted Epitopes

In these analyses, allergic and non-antigenic epitopes were removed, and the remaining probable antigenic and non-allergic epitopes were selected for further analysis. The solubility was determined via InvivoGen and ToxinPred for toxicity to avoid the toxic effect of all the toxic epitopes. Ten epitopes were found to be soluble and non-toxic and were considered for the designing of multi-epitopes as mentioned in Figure 3. The selected epitopes fulfilling all the epitopes parameters are tabulated in Table 3. The prediction software assigns an ‘antigenicity value’ to each antigenic proteins, which correlates with their ability to stimulate immune responses. Higher values determine the strong antigenic potential of a protein, and vice versa. A value >0.4 was considered appropriate to identify proteins likely able to induce strong immunological responses [30]. The allergenicity, solubility and non-toxicity results generated by the software are categorical and in a Yes/No fashion.

### 3.8. Multi-Epitopes Vaccine Construct

The multi-epitopes based vaccine construct consisted of different epitopes rather than a single epitope in order to generate strong and protective immune responses. The multi-epitopes vaccine was designed by linking all top 10 screen epitopes with each other with the help of GPGPG linkers. Furthermore, the designed vaccine was also linked with Cholera Toxin B subunit adjuvant via an EAAAK linker to enhance the efficacy of immune response [39], which helps in the stability of the construct. Using the Protparam tool, the physiochemical properties of the construct were checked. This would help experimental vaccinology in the formulation of the vaccine. The designed multi-epitopes vaccine is schematically given in Figure 4. Revalidation on the antigenicity, allergenicity and toxicity of the design vaccine sequence was achieved using the same software used for epitopes evaluation and as described in the methods section. The designed vaccine was found to be antigenic (0.87), non-allergic and non-toxic, thus further augmenting the proposed vaccine model as a good vaccine candidate.

#### D Structure of Vaccine, Loop Modeling, and Refinement

Using a 3Dpro SCRATCH predictor tool, the tertiary structure of the multi-epitopes construct was modeled as shown in Figure 5. The model vaccine 3D structure is the best we can get and by the best ab initio structure modeling algorithm. As the template structure is absent in PDB, we only rely on the ab initio algorithm to get the best possible 3D model. All of the following loops were modeled: Met1-Lys5,Ala17-Gly21-Cys30,Ile38-Glu50-Ile60-Ile61-Pro74-Glu100-Asn111,Gly130-Ser134,Phe138-Glu149,Gly153-Pro170,Gly181,Arg200,Leu201-Asn220-Pro221, Gly241,Gly242-Gly253,Gly254-Leu264, After loops modeling, the modeled structure was subjected for refinement in galaxy web services for refining 2 [40].

### 3.9. Disulfide Engineering and Codon Optimization

To avoid the breakdown of the designed vaccine weak regions, disulfide engineering was done to stabilize the bonding between residues having unfavorable energy [41]. Disulfide bonds were established for residue pairs that were sensitive to enzymatic breakdown (non-favorable energy), as shown by yellow sticks in the mutated structure given in Figure 6 and the amino acids residues are tabulated in Table 4. The residue pairs with an energy value of >1 kcal/mol were highlighted by Design 2.0, which can be used for establishing disulfide bonds. These amino acid pairs have high unfavorable energy and are not stable.

After the above process, through the use of the Java Codon Adaptation Tool (JCat), the sequence of the designed vaccine construct was first reverse translated to DNA sequence to get the maximum level of expression of vaccine in the *E. coli* vector and calculate it with the aid of a codon adaptation index (CAI) and its GC percentage values. The designed vaccine was then expressed into the pET-28a (+) vector through SnapGene, as represented in Figure 7.

### 3.10. Molecular Docking

Blind docking was performed for the prediction of interaction between the vaccine with MHC-I (PDB ID: 1L1Y), MHC-II (1KG0), and TLR-4 (PDB: 4G8A) with the help of PATCHDOCK [43] and the predicted top 20 docked complexes. The results obtained from the PatchDock server are tabulated in Appendix A. The docked solutions with lowest global energy were considered the most stable and were subjected to further investigation.

### 3.11. Refinement of Docked Complexes

The FIREDOCK [53] server re-ranked the solution after removing many steric clashes and intermolecular conformational errors. Solution 10 in the case of the vaccine construct with MHC-I was selected as having the lowest global energy (7.65 kJ·mol^−1^), attractive van der Waals (0 kJ·mol^−1^), repulsive VdW (0 kJ·mol^−1^), atomic contact energy (ACE) (0 kJ·mol^−1^) and hydrogen bonding (0 kJ·mol^−1^). In the case of MHC-II solution, number 8 has the lowest global energy (3.11 kJ·mol^−1^), attractive van der Waals (−4.44 kJ·mol^−1^), repulsive van der Waals (1.75 kJ·mol^−1^), ACE (1.71 kJ·mol^−1^) and hydrogen bond energy (0. kJ·mol^−1^). In case of TLR-4, solution number 4 was selected as having the lowest global energy (−15.33 kJ·mol^−1^), attractive van der Waals (−22.6), repulsive VdW (7.55), ACE) (2.29 and hydrogen bonding (−2.62 kJ·mol^−1^) were found more stable due to lower global energy and selected for high structure configuration studies through UCSF Chimera 1.13.1 [45]. Docking scores of the refine docked complexes of MHC-I, MHC-II and TLR-4 are mentioned in Appendix A, respectively. The 3D docked complexes are presented in Figure 8A–C.

### 3.12. Residues Wise Interaction Analysis of MHC-MHC- and TLR-4 to Vaccine

Peptide antigen processing and presentation to immune cells by MHC molecules is crucial for the adaptive immune response. Before antigen processing and presentation, the foreign peptide antigen is required to interact with different types of immune cells to generate an appropriate immune response. These intermolecular interactions of MHC-I, MHC-II and TLR-4 are critical to deciphering residues important from a vaccine recognition perspective. The model vaccine construct showed strong interactions with several key amino acid residues of MHC-I, MHC-II and TLR-4 immune cells receptor molecules as find out in UCSF chimera and tabulated in Table 5. The shortlisting of the interactions shown in Table 5 is done based on bond distance. The majority of these interactions are within 5 Å.

### 3.13. Molecular Dynamic Simulation

Molecular dynamic simulation is a computer simulation process for the analysis of the dynamic behavior of macromolecules. Molecular dynamics simulations of docked complexes were performed for 200 nanoseconds (ns) to evaluate the structural stability of the systems. The simulations were carried out using the AMBER20 simulation package [46]. The analysis consists of root mean square fluctuation (RMSF) and radius of gyration (RoG), and root mean square deviation (RMSD). The RMSD graph plot is constant with no major structural changes observed. RMSF was observed by the residue flexibility of the receptors in the presence of the vaccine molecule. The majority of systems residues are within a good stability range (<3 Å). The RoG analysis was calculated to examine the system compactness versus time, and it was concluded that there are no drastic changes that occur in all systems. Graphical representations of RMSD, RMSF, and RoG are presented in Figure 9A–C.

### 3.14. Estimation of Binding Free Energies of Vaccine Construct with MHC-I, MHC-II, and TLR-4

Through the MMPBSA.py module, the MMPBSA/MM/GBSA binding free energies of the vaccine- receptor was estimated [54]. Only 100 frames were considered while estimating binding free energies. The total binding free energy of a vaccine with TLR-4, MHC-I, and MHC-II were −112.14 kcal/mol, −92.26 kcal/mol, and −89.1 kcal/mol, respectively, as given in Table 6. The net binding energy contribution from van der Waals energy and electrostatic (hydrogen bonding) parameters were the most favorable. Moreover, the insignificant energy involvement from the polar salvation was noted.

### 3.15. Vaccine Immune Simulation

The immunogenic efficacy of the final vaccine construct was evaluated by performing in silico immune simulations with the help of the C-immSim server 10.1 for 350 days [50]. The humoral immune response to the vaccine antigen was dominated by IgG and IgM antibodies. The innate immune response generated by the vaccine construct was observed in the form of IgM antibodies. The secondary immune response followed by other immune responses also leads to a maximum level of production of B-cell and IgM, IgG, IgM, IgG1 + IgG2, IgG1, and IgG2, as mentioned in Figure 10A. Similarly, interferon-γ production in response to the antigen was also observed in a titer of 400,000 for 35 days as mentioned in Figure 10B. An increase in other types of immune response T_c_ (cytotoxic killer T-cell), macrophages (Mφ), natural killer cells, and dendritic and epithelial cells is shown in Appendix A.

## 4. Discussion

AR is the outcome of the bacterial evolution process to make itself resistant to antibiotics. To prevent infection caused by AR pathogens, vaccination is an alternative approach to generate a proper immunological response against specific organisms [55]. *P. shigelloides* is one of the AR bacterial species and shows resistance to several classes of commercially available antibiotics such as azithromycin, penicillin, doxycycline, and erythromycin. *P. shigelloides* is a group of opportunistic gram-negative, motile, and rod-shaped pathogens belonging to the *Enterobacteriaceae* family. It causes many infections, including diarrhea, gastrointestinal infection, CNS abnormalities, neonatal sepsis and vision problems. *P. shigelloides* isolates from hospital patients are reported to show high resistance to antibiotics leading to high mortality and morbidity rates (https://doi.org/10.3389/fmicb.2018.03077 (accessed on 24 September 2022)).

Hence, in this study, we designed an in silico vaccine model against *P. shigelloides* to lower the burden of AR [56]. The genomics revolution greatly helped in designing novel therapeutic and prophylactic vaccine candidates for traditional vaccine development. Next-generation sequencing of bacterial pathogens and advanced bioinformatics practices in vaccinology are now commonly employed for the identification of putative surface-associated antigens [57]. RV is a safe, specific, and potent approach and is used to identify putative surface-associated proteins without the need to culture the microorganisms [10]. By using the RV approach, the meningococcal serogroup B (4CMenB) vaccine was effectively developed [58]. The method has been used for other bacterial and viral pathogens as well. Examples are the Crimean-Congo hemorrhagic fever virus (https://doi.org/10.1038/s41598-022-12651-1 (accessed on 24 September 2022)), *Onchocerca volvulus* (https://doi.org/10.3389/fitd.2022.1046522 (accessed on 24 September 2022)), and *Listeria monocytogenes* (https://doi.org/10.1038/s41467-022-33721-y (accessed on 24 September 2022)). Traditional vaccinology is a failure for pathogens that are unable to be cultured or grown in vitro. As compared to conventional RV, pan-genomic reverse vaccinology (PGRV) is more effective as it screens highly conserved targets than strain specific ones. For example, the genome of *Streptococcus agalactiae* determined four protective antigens identified with the help of the PGRV approach [1]. Traditional vaccinology is costly and time-consuming and in high need of human resources. We use a novel therapeutics RV approach in combination with biophysical approaches to design a multi-epitopes based vaccine against *P. shigelloides*.

A good vaccine candidate has the following properties: it should be antigenic, immunogenic, non-homologous, non-allergic, and is located on the pathogen surface region. All of these properties are literature-based and highly desirable to design a chimeric vaccine against a specific pathogen. Immune cell epitopes prediction, analysis and processing of potential and safe antigens, population coverage and conservation analysis, toxicity prediction of the antigens, allergenicity evaluation, docking and simulation approaches, and binding energies estimations are steps used in computational vaccinology [59]. In the current study, the whole genome of bacteria was retrieved from the NCBI. The core genome contains the sequences present among all strains. In the core proteome, we selected only those protein sequences showing non-redundancy, essential for pathogen survival, non-homology to the human, and normal microbiota [22]. The redundant genome was discarded because of double sequence representation. The outer membrane, extracellular, and periplasmic proteins were selected because they are well exposed to the environment and have great potential to provoke an immune response. A homology analysis of the subcellular localized proteins was performed against humans and three normal microbiota of the human to avoid autoimmune responses due to similarity between human and microbiota species.

Three different types of subcellular localized proteins: (i) flagellar hook protein FlgE (ii) hypothetical protein; and (iii) hemoglobin/transferrin/lactoferrin family receptor) were selected due to non-allergic, probable antigenic, and non-similar with human and microbiota. The shortlisted proteins were utilized for B and T-cell epitopes prediction [37]. The predicted B and T–cell epitopes mainly stimulate/ boost up both humoral and cellular immune responses. T-cell epitopes prediction involves the evaluation of B-cell epitopes for their effective binding with molecules of both classes MHC-I and MHC-II alleles. At last, 10 probable antigenic, non-toxic, non-allergic, and good water soluble epitopes were shortlisted for multi-epitopes.

In the multi-epitopes designing phase, all the shortlisted epitopes were linked by GPGPG linkers to design a multi-epitopes vaccine construct. The designed vaccine construct was further linked with Cholera Toxin B subunit adjuvant via EAAAK linker for making the designed vaccine more efficient. The tertiary structure of the vaccine construct was modeled and refined to maintain structural stability. The construct was subjected to blinding docking to check the binding interaction between the construct and MHC-I, MHC-II, and TLR-4, and to examine the immune responses. We found solution 10 in MHC-I, 8 in MHC-II, and 4 in TLR-4 more stable because of lower global energy. The behavior of molecules within the host cell was achieved in the molecular dynamic phase and binding energies of a construct with receptor were estimated. The integrated SP and RV approach successfully identified an antigenic epitope for the design of a chimeric vaccine to boost up the host immune response against *P. shigelloides*.

Clinically, the designed vaccine may be quite useful as the vaccine contains core epitopes and thus could provide broad spectrum immune protection against all strains of the pathogen. Also, the vaccine is safe as it is non-allergic and non-toxic. Some limitations of the study need to be overcome in future studies. First, the need for an experimental evaluation to get the best combination epitopes in the vaccine construct for maximum level of immune responses must be done. Second, the refinement of MHC molecules epitopes prediction algorithms is under way. Lastly, the real immune protection of the vaccine required extensive in vivo and in vitro testing.

## 5. Conclusions

In the current study, we employed RV, SP, and immunoinformatics approaches to design a multi-antigenic epitope- based vaccine against *P. shigelloides*, which is one of the most troublesome human pathogens and is highly resistant to several antibiotics. This is worrisome in addition to the absence of a licensed vaccine against the pathogen. By using core, non-homology, non-redundant proteins, we designed a vaccine consisting of non-allergic, antigenic, non-toxic, and soluble epitopes. The epitopes were joined to each other by GPGPG linkers and linked with cholera toxin B subunit adjuvant with the help of another EAAAK linker to enhance the potency of the designed vaccine. The vaccine candidate was used as a comprehensive immune system inducer and the strongest candidates were prioritized in future vaccine development efforts to prevent future *P. shigelloides* disease outbreaks. We believe that the product of this work is a designed peptide vaccine for researchers to investigate its immune protection ability in vivo, and the findings of the study will increase the vaccine antigens library against *P. shigelloides* as well as fast-track the vaccine development process. Furthermore, as the vaccine design is based on proteins that form the core genome of the pathogen, the vaccine is likely to provide cross-protection against all sequenced strains of the bacteria.

## Figures and Tables

**Figure 1 vaccines-10-01886-f001:**
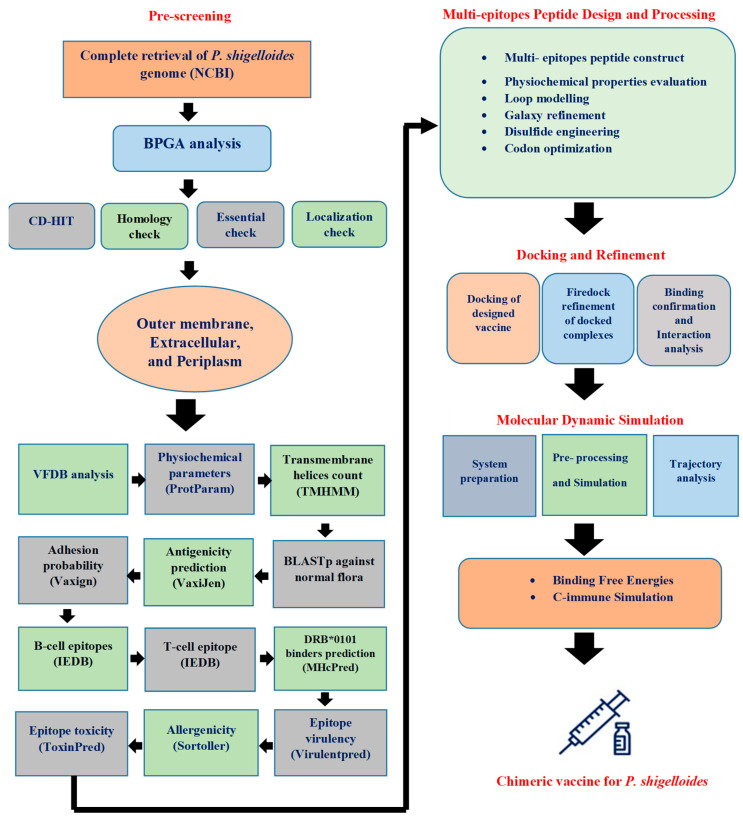
Methodology flow for designing a novel multi-antigenic epitopes vaccine against *P. shigelloides*.

**Figure 2 vaccines-10-01886-f002:**
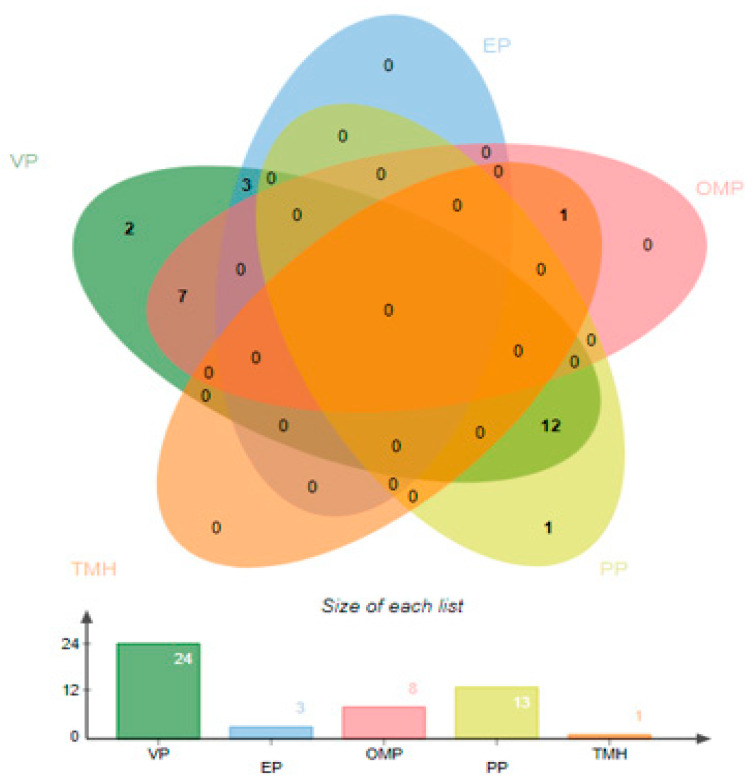
The number of surface localized proteins obtained is represented as Venn. Number of virulent proteins (VP), periplasmic proteins (PP), outer membrane proteins (OMP), extracellular proteins (EP), and transmembrane helices (TMH).

**Figure 3 vaccines-10-01886-f003:**
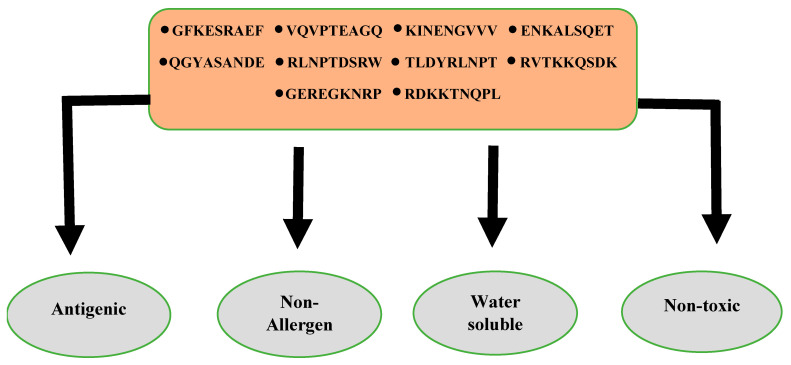
Final 10 shortlisted epitopes used for construction of a multi-epitopes-based vaccine.

**Figure 4 vaccines-10-01886-f004:**
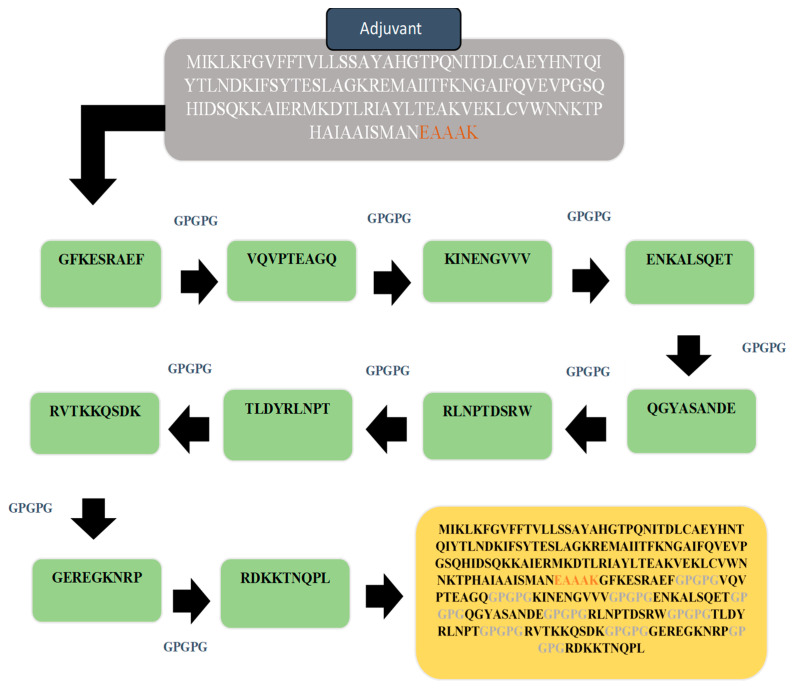
Schematic diagram of 264 amino acid long vaccine construct sequence. The selected epitopes are represented in green color boxes, GPGPG linkers for linking epitopes with each other while EAAAK linker to join the designed vaccine construct with cholera toxin B subunit adjuvant as shown in the purple color box.

**Figure 5 vaccines-10-01886-f005:**
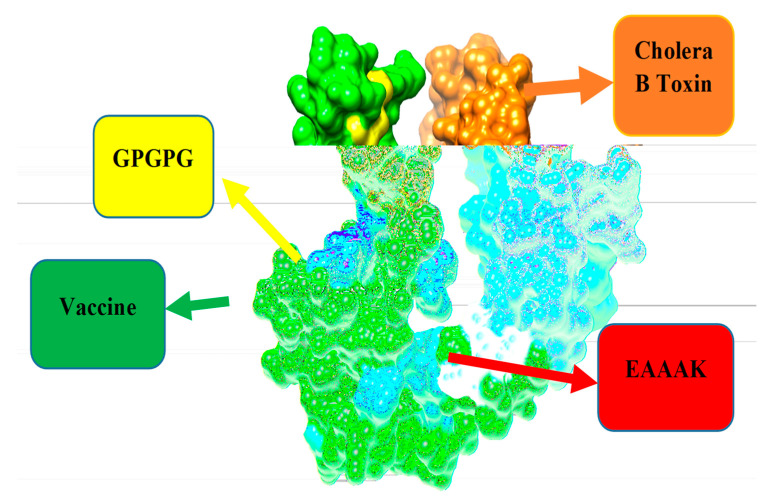
3D structure of chimeric multi-epitopes vaccine construct.

**Figure 6 vaccines-10-01886-f006:**
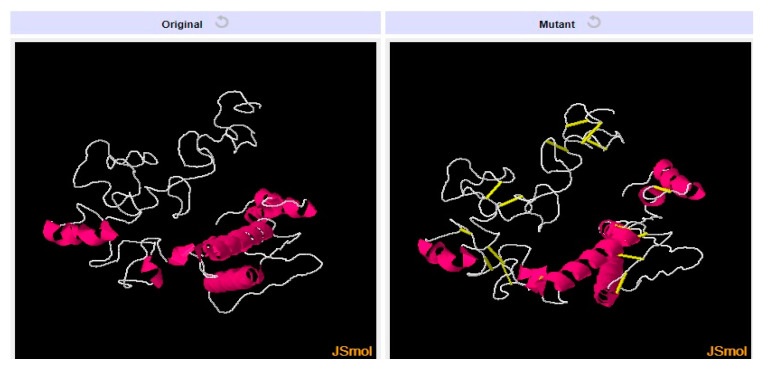
Left side structure represents original structure while right side structure represents the mutated structure of final vaccine construct.

**Figure 7 vaccines-10-01886-f007:**
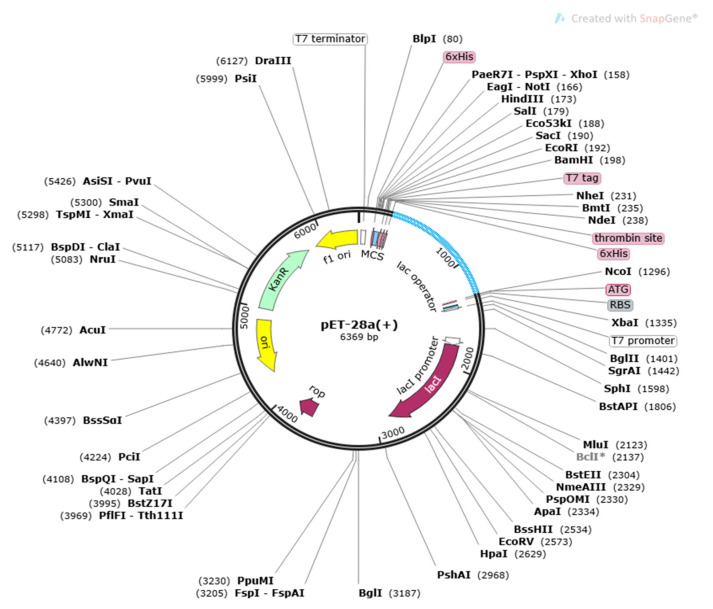
Cloning of vaccine construct into *E. coli* k12 strains pET28a vector.

**Figure 8 vaccines-10-01886-f008:**
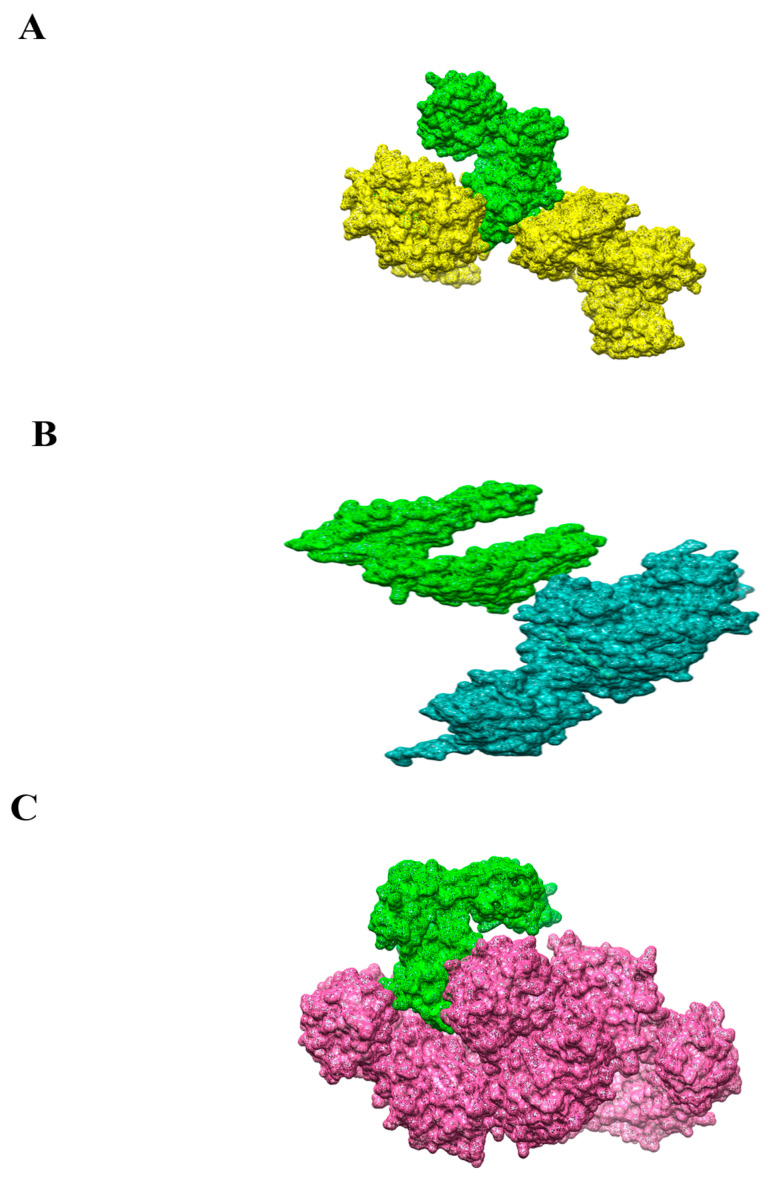
(**A**) 3D structure of the docked complex of chimeric vaccine (green mesh) and receptor MHC-I (yellow mesh). (**B**). 3D structure of the docked complex of chimeric vaccine (green mesh) and receptor MHC-I (dodger blue mesh). (**C**). 3D structure of the docked complex of chimeric vaccine (green mesh) and receptor MHC-I (magenta mesh).

**Figure 9 vaccines-10-01886-f009:**
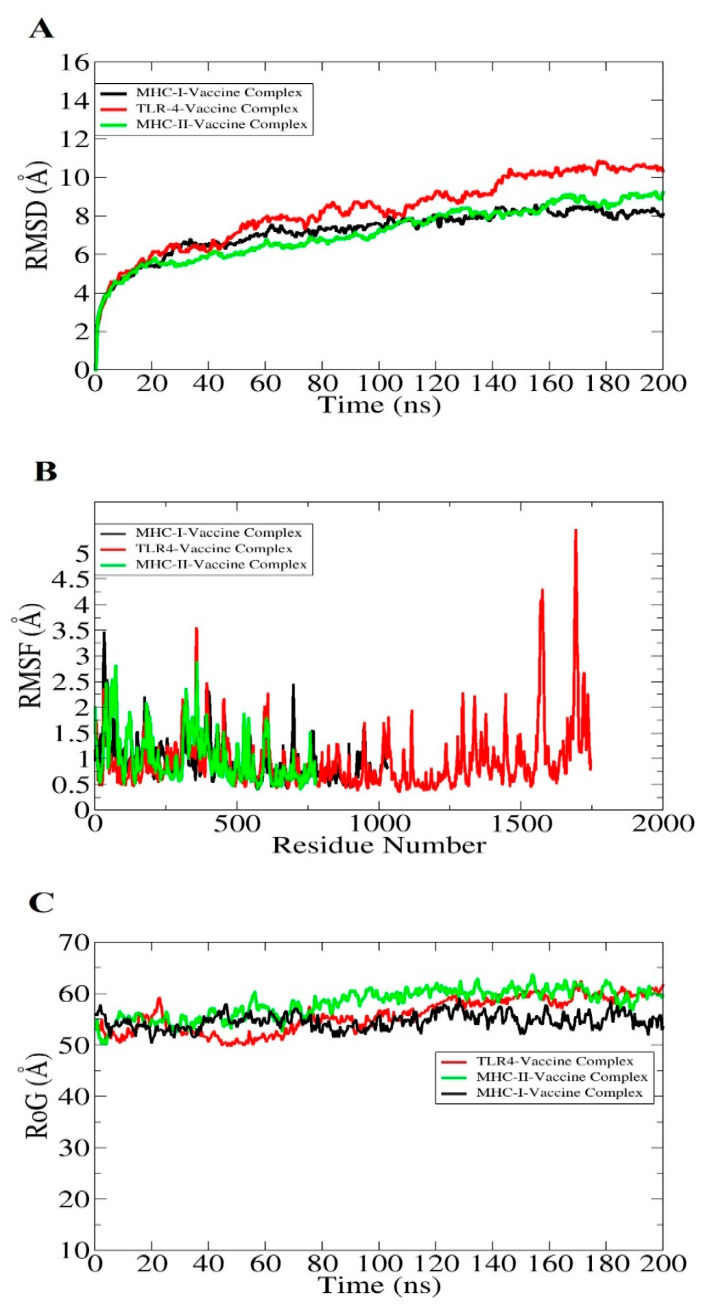
Statistical analysis of simulation trajectories; three output values are depicted here: RMSD (**A**), RMSF (**B**), and RoG (**C**).

**Figure 10 vaccines-10-01886-f010:**
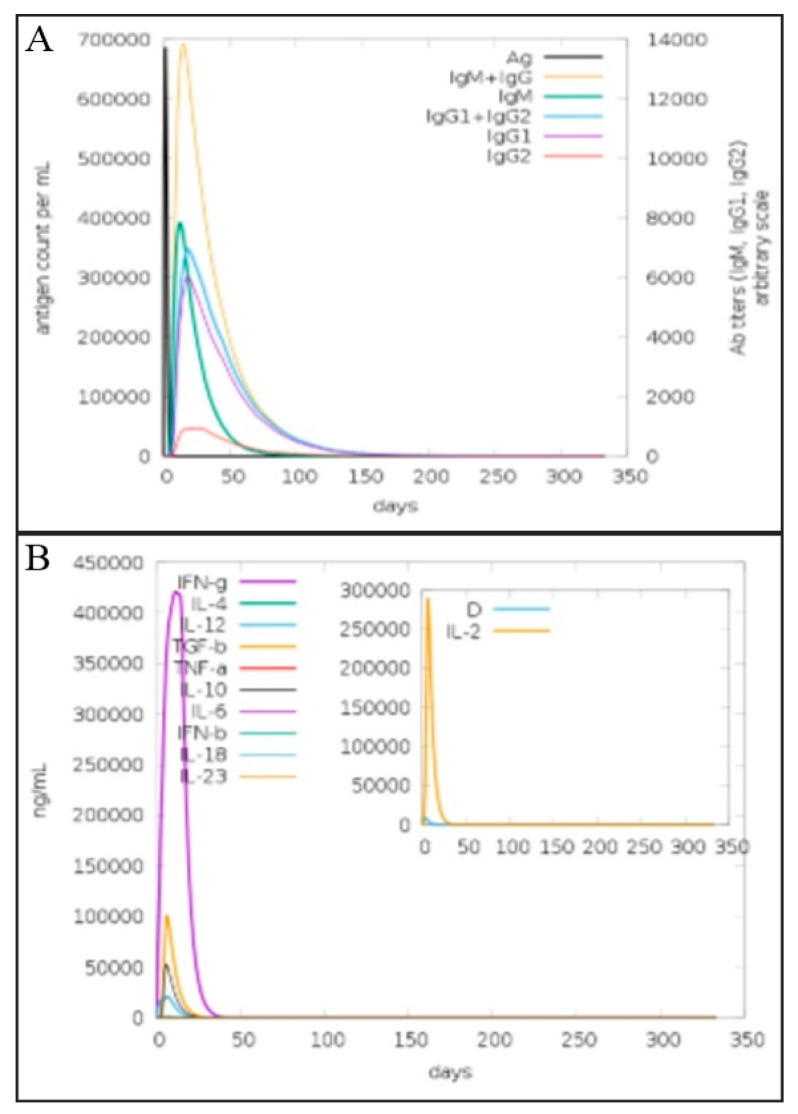
(**A**). Immunoglobulin titer (different color peak) in response to multi-epitopes vaccine injection (black color peak). (**B**). Elicitation of interleukins level after injection of multi-epitopes vaccine construct as represented by C- immune simulation analysis.

**Table 1 vaccines-10-01886-t001:** Size and GC content of *P. shigelloides* genomes.

Strain	Size (Mb)	GC%
MS-17-188	3.97036	51.4817
NCTC10360	3.40598	52

**Table 2 vaccines-10-01886-t002:** Homology check of protein with human genome and normal flora, antigenicity, allergenicity, and adhesion probability analysis.

Protein	Gene Ontology	Human	*Lactobacillus rhamnosus*	*Lactobacillus casei*	*Lactobacillus johnsonii*	Antigenicity
Flagellar hook protein FlgE	Bacterial-type flagellum basal body	No-Similarity	No Similarity			0.82
Hypothetical protein	Membrane protein					0.70
Hemoglobin/transferrin/ lactoferrin family receptor	Integral component of membrane					0.70

**Table 3 vaccines-10-01886-t003:** Determination of antigenicity, allergenicity, solubility, and toxicity of predicted epitopes.

MHcPred	Antigenicity	Allergenicity	Solubility	ToxinPred
GFKESRAEF	0.52	Non-Allergen	Soluble	Non-Toxin
VQVPTEAGQ	0.50			
KINENGVVV	0.77			
ENKALSQET	0.70			
QGYASANDE	0.70			
RLNPTDSRW	1.28			
TLDYRLNPT	2.23			
RVTKKQSDK	1.49			
GEREGKNRP	2.24			
RDKKTNQPL	1.19			

**Table 4 vaccines-10-01886-t004:** Disulfide engineering of the designed vaccine. Sequence number (S.N), amino acid (A.A) Chi3, Energy and Sum B- factors of mutated amino residues.

S.N	A.A	S.N	A.A	Chi3	Energy	Sum B-Factors
11	Thr	29	Leu	90.87	5.4	0
19	Ala	22	Thr	68.11	4.56	0
19	Thr	41	Leu	89.79	2.9	0
38	Pro	77	Gln	115.92	3.87	0
74	Glu	116	Ala	80.19	3.86	0
32	Trp	112	Lys	79.07	1.07	0
104	Lys	135	Arg	103.02	0.99	0

**Table 5 vaccines-10-01886-t005:** Residues wise interaction of vaccine construct to receptors.

Vaccine Complex	Interactive Residues
MHC-I	Ala128,Asn24,His145,Phe131,Ala149,Asp106,Ile52,Pro20,Ala136,Asp223,Leu 272,Ser132,Ala150,Glu148,Leu201,Thr80,Arg157,Gly 104,Lys 19,Trp 167,Arg 75,Gln 141,Met 4,Trp 51,Arg 169,Glu 16,Met 99
MHC-II	Arg256,Gln77,Leu219,Ser191,Asn10,Gln197,Lys84,Thr4,Asn124,Gly25,Lys232,Thr230,Asn192,Gly197,Met122,Trp109,Ala19,His20,Phe09,Trp208,Ala191, His74,Pro18,Tyr188,Asp 43,His77,Pro224, Val 108,Gln 02,Ile45,Pro 238,Val 164
TLR-4	Arg355,Gln 145,Ile38,Met 1,Ser312,Ala128,Gln152,Ile454,Met58,Ser569,Asn 65,Glu50,Lys3,Phe 06,Thr27,Asn544,Glu137,Val524,Phe09,Thr 260,Asp 100,Glu 161,Lys 44,Phe 396,Val 73,Asp194,Gly 153,Lys55,Phe500,Val122,Cys 40,His115,Lys64,Pro140,Val165,Gln70,Ile36,Lys247,Pro23,Val 146

**Table 6 vaccines-10-01886-t006:** MMGBSA/ PBSA binding free energies of the design vaccine with receptors.

Energy Parameter	TLR-4-Vaccine Complex	MHC-I-Vaccine Complex	MHC-II-Vaccine Complex
MM-GBSA			
VDWAALS	−75.06	−69.84	−76.32
EEL	−66.75	−56.06	−45.25
Delta G gas	−141.81	−125.9	−121.57
Delta G solv	29.67	33.64	32.47
Delta Total	−112.14	−92.26	−89.1
MM-PBSA			
VDWAALS	−75.06	−69.84	−76.32
EEL	−66.75	−56.06	−45.25
Delta G gas	−141.81	−125.9	−121.57
Delta G solv	28.99	30.34	28.14
Delta Total	−112.82	−95.56	−93.43

## Data Availability

The data presented in this study are available within the article or in Appendix A.

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
