# Peer review of "Computational Design of a Chimeric Vaccine against Plesiomonas shigelloides Using Pan-Genome and Reverse Vaccinology"

_vaccines, 2022, doi:10.3390/vaccines10111886_

Round 1

Reviewer 1 Report

The topic addressed by the manuscript by Sajjad Ahmad and coworkers falls within the scope of the journal and the results are of interest to its readers. The project design and results discussion are both not completely sound and the conclusions are partially supported by the data presented by the authors.

There are several, severe issues with this work:

1) Analysis on antigenicity, allergenicity, toxicity, etc. have been performed on starting proteins, epitopes, etc., not on the long peptide sequence the authors suggest as a possible vaccine. It is well known that the above mentioned properties are strongly affected by the sequence and 3D-structure of the peptide: it is VERY unlikely that the antigenicity, allergenicity, toxicity, etc., profile of the short peptides is maintained in the long sequence proposed.

2) It seems that the authors modeled also the 3D-structure of the adjuvant portion. The cholera toxin b subunit 3D-structure has been solved by X-Ray diffraction analysis. The authors should ensure that the native 3D-structure is preserved in their modeling.

3) More importantly, the docking is not convincing. The vaccine seems not to interact with MHC-I, MHC-II, or TLR-4 effectively. Line 390: "The model vaccine construct showed strong interactions with several key amino acid residues". This claim is not supported by the data presented by the authors. Table 5 does not provide any information.

4) The authors should unambiguously state the effect of the multi-epitope peptide on the interactions of the proposed construct with the receptors - for instance, over the cholera toxin b subunit alone. Are there more interactions when the proposed peptide sequence is present?

In conclusion, the paper can be considered further only if the authors:

i. Perform the antigenicity, allergenicity, toxicity, etc. analysis on the whole vaccine construct. [By the way, the authors themselves state that (lines 459-463)].

ii. Report the interactions in the docked complexes (with each of the three receptors considered) in terms of number of hydrogen bonds, salt bridges, etc. A figure illustrating the contact sites should be included.

iii. Double-check the 3D-structure of the cholera toxin b subunit obtained by their modeling against its deposited X-ray diffraction structure.

iv. Show that the construct is more effective in the interaction with the chosen receptors than the adjuvant alone.

The authors may consider the following paper as a possible model for their revision: Omoniyi, A.A., Adebisi, S.S., Musa, S.A. et al. In silico design and analyses of a multi-epitope vaccine against Crimean-Congo hemorrhagic fever virus through reverse vaccinology and immunoinformatics approaches. Sci Rep 12, 8736 (2022). https://doi.org/10.1038/s41598-022-12651-1

Line 504: "The designed vaccine has a binding affinity with immune receptors". The statement is too generic and not supported by data. Predicting binding affinity from structural models, for instance in terms of dissociation constant, is hampered by the complexity of the interactions, particularly for flexible complexes. The statement should be removed if no quantification can be provided.

OTHER POINTS, to be addressed after the above questions have been fully clarified:

Introduction should be rewritten: unnecessary information should be deleted (e.g., from lines 56-86: for examples see below "minor points"). A line on why Plesiomonas shigelloides is considered an AR bacterial species and a brief discussion on the state-of-the-art of vaccines against Plesiomonas shigelloides, and Shigella, should instead be included (by the way, line 439 needs a reference, as well). The following papers should be considered:

- Michael Janda J., Abbott S.L., McIver C.J., Plesiomonas shigelloides revisited, (2016) Clinical Microbiology Reviews, 29 (2), pp. 349 - 374 DOI: 10.1128/CMR.00103-15

- Selim, S.; Almuhayawi, M.S.; Zakai, S.A.; Salama, A.A.; Warrad, M. Distinction between Antimicrobial Resistance and Putative Virulence Genes Characterization in Plesiomonas shigelloides Isolated from Different Sources. Antibiotics 2022, 11, 85. https://doi.org/10.3390/antibiotics11010085

- Taylor DN, Trofa AC, Sadoff J, Chu C, Bryla D, Shiloach J, Cohen D, Ashkenazi S, Lerman Y, Egan W, et al. Synthesis, characterization, and clinical evaluation of conjugate vaccines composed of the O-specific polysaccharides of Shigella dysenteriae type 1, Shigella flexneri type 2a, and Shigella sonnei (Plesiomonas shigelloides) bound to bacterial toxoids. Infect Immun. 1993 Sep;61(9):3678-87. doi: 10.1128/iai.61.9.3678-3687.1993

- Louis-Antoine Barel & Laurence A. Mulard (2019) Classical and novel strategies to develop a Shigella glycoconjugate vaccine: from concept to efficacy in human, Human Vaccines & Immunotherapeutics, 15:6, 1338-1356, DOI: 10.1080/21645515.2019.1606972

- Herrera, C.M.; Schmitt, J.S.; Chowdhry, E.I.; Riddle, M.S. From Kiyoshi Shiga to Present-Day Shigella Vaccines: A Historical Narrative Review. Vaccines 2022, 10, 645. https://doi.org/10.3390/vaccines10050645

Other approaches should also be briefly mentioned, such as:

Chunjun Qin, Benjamin Schumann, Xiaopeng Zou, Claney L. Pereira, Guangzong Tian, Jing Hu, Peter H. Seeberger, and Jian Yin, Total Synthesis of a Densely Functionalized Plesiomonas shigelloides Serotype 51 Aminoglycoside Trisaccharide Antigen, Journal of the American Chemical Society 2018 140 (8), 3120-3127, DOI: 10.1021/jacs.8b00148

Line 104: please, replace "ready-to-use" with "designed".

Par 2.4 Cd-hit analysis. The statement "sequence identify (identity) of < 30 % and bit score of > 100" should be briefly explained to non-specialists. Actually, this par. should be removed and merged with lines 146-148 of par. 2.6. By the way, on the identification of the virulence factors of P. shigelloides (with reference to the statement of lines 251-253) see: Ekundayo TC, Okoh AI. Pathogenomics of Virulence Traits of Plesiomonas shigelloides That Were Deemed Inconclusive by Traditional Experimental Approaches. Front Microbiol. 2018 Dec 21;9:3077. doi: 10.3389/fmicb.2018.03077.

Figure 2, 3 and 4 are not useful and should be either removed or moved to the Supporting information. Paragraphs 3.1, 3.2, and 3.3 can be merged into one.

Lines 262-263: "Those proteins were selected showing molecular properties less than 110 KD and instability index less than 40" The readers might not be specialists of the ExPASy server parameters. Significance of the mentioned values must be explained.

Table 2. The values reported for antigenicity (by the way, those from allergenicity and adhesion probability analysis perhaps are missing) should be explained. Lines 274-275 state: "The three proteins were found antigenic, non-allergen and showing adhesion probability value greater than 0.6 indicating good vaccine candidates". I wonder what the values reported in Table 2 refer to? The reasons of choosing that threshold value (0.6) must be explained (the software used for the analysis should be named). The same holds true for Table 3 (how do the reported antigenicity values prove that the selected peptides are good antigens? The reported numbers are not enough) and for the "score 0.8" reported in Par. 3.8, line 283. Please, explain why 0.8.

Line 340: "In multi-peptide base vaccine constructs, some of the peptide residues are enzyme degradable". Proteolytic degradation affects peptide bonds, not single residues that, if anything, may be recognized, not 'degraded'.

The statement (line 343): "All the residues which are sensitive to enzymatic degradation are replaced with cysteine bond" is meaningless. What do the authors mean for "residues sensitive to enzymatic degradation"? What do the authors mean with "cysteine bond" Could that be that two residues are replaced by a cystine (Cys-S-S-Cys)? Please, use the correct terms.

Table 4: The reported values do not help in understanding which residues were actually replaced and why. Sequence number (S.N)? Chi3? Energy? Sum B- factors? The epitope sequences were never numbered in the text. Some residues are apparently making disulfide bonds with two different residues at the same time (what is the difference between "41 Leu" and "41 Leu", third and fourth rows?) The rationale underpinning the chosen substitutions is not clear and should be discussed.

Line 362: "Those solutions were subjected further based on global energy" subjected? Statement not clear.

Line 420: "A proper increase in the production of adaptive immune response". The statement is too generic.

Lines 425-427: "Similarly, production of several types of interferon-gamma was also observed which greater than 400000 is for almost 35 days was as mentioned in Figure 13.B." ? Statement not clear.

Discussion: this session should be more concise.

MINOR POINTS:

Abstract, line 37: "In total of 24 virulent proteins, 03 proteins; flagellar hook.." (?). Statement not clear. Line 40: "[...] and solubility analysis." Please, remove "analysis". line 47, the abbreviation MHC (major histocompatibility complex) should be explained when first encountered in the text.

Introduction, lines 62-71: "About ... respectively". The statement needs a reference. Lines 65-71: "For combating bacterial diseases [...] Boosting host immunity [...] For combating bacterial diseases [...] boost up a human immune response". The same concept - which is by itself not useful to the manuscript - is repeated multiple times. "At present, there is no licensed vaccines are present". Please, correct. Line 74-75: "A vaccine is a biological preparation of weakened, killed, or fragmented microbes that provide active acquired immunity against infectious diseases" This is not the accepted definition of vaccine. Please correct with something like: ": A vaccine is a preparation used to stimulate the body's immune response against diseases. It can be made up of weakened, killed, or fragmented microbes." or, better, remove the statement. Lines 76, 77 and 79: "A study reported by WHO claims that vaccines can prevent 2.5 million deaths per day worldwide."; "A recent epidemiological study..."; "Vaccine saves five lives per minute worldwide." All of those statements should be removed since they do not ground the present study. If maintained, they need supporting reference(s).

Table 1. Why the GC content values of the two genomes differ so much in the significant figures?

Lines 285-299 should be moved to the Experimental session.

Line 303. Please, remove "analysis".

Line 341: "To avoid this problem, disulfide engineering of vaccine construct was for disulfide engineering to supplement the degradable prone regions with disulfide bonds". Please rephrase this statement.

Lines 442-444: "P. shigelloides in rare cause’s neonatal sepsis and in hospital settings which shows resistance to antibiotics with high mortality and morbidity rate" ? Statement not clear.

SOME OF THE TYPOS: Line 39 "receptor protein" (not proteins); Line 90 "it screens"; line 97 "mainly causes"; line 122 "that shows sequence..."; Line 384 "to interact" (remove "be"); Line 438: "shows"; Line 454: "is more effective as it screens" (by the way, remove "in specific"). Several more are present in the manuscript.

Author Response

Response to Reviewer Comments

We thank the Referee for spending time and interest in our work and for helpful comments that will greatly improve the manuscript. We have checked all the general and specific comments provided by the Referee and have made all the necessary changes according to his indications. Please refer to yellow highlighted sections in the revised manuscript.

Reviewer # 1

Comments and Suggestions for Authors

The topic addressed by the manuscript by Sajjad Ahmad and coworkers falls within the scope of the journal and the results are of interest to its readers. The project design and results discussion are both not completely sound and the conclusions are partially supported by the data presented by the authors.

There are several, severe issues with this work:

1) Analysis on antigenicity, allergenicity, toxicity, etc. have been performed on starting proteins, epitopes, etc., not on the long peptide sequence the authors suggest as a possible vaccine. It is well known that the above mentioned properties are strongly affected by the sequence and 3D-structure of the peptide: it is VERY unlikely that the antigenicity, allergenicity, toxicity, etc., profile of the short peptides is maintained in the long sequence proposed.

Response: Thank you for the valid point. The mentioned analysis are done in for the designed vaccine and added to section 3.10 of the revised manuscript.

2) It seems that the authors modeled also the 3D-structure of the adjuvant portion. The cholera toxin b subunit 3D-structure has been solved by X-Ray diffraction analysis. The authors should ensure that the native 3D-structure is preserved in their modeling.

Response: Thank you for the comment. The model vaccine 3D structure is the best we can get and by the best ab initio structure modeling algorithm. As the template structure is absent in PDB, we only relay on ab initio algorithm to get the best possible 3D model.The text is added to revised section 3.11 of the revised manuscript.   

3) More importantly, the docking is not convincing. The vaccine seems not to interact with MHC-I, MHC-II, or TLR-4 effectively. Line 390: "The model vaccine construct showed strong interactions with several key amino acid residues". This claim is not supported by the data presented by the authors. Table 5 does not provide any information.

Response: Thank you for comment. It is very clear that the interaction is strong between vaccine and receptors. Table 5 contains all the interacting residues of the receptors with the vaccine.

4) The authors should unambiguously state the effect of the multi-epitope peptide on the interactions of the proposed construct with the receptors - for instance, over the cholera toxin b subunit alone. Are there more interactions when the proposed peptide sequence is present?

Response: The comment is already addressed above.

In conclusion, the paper can be considered further only if the authors:

  1. Perform the antigenicity, allergenicity, toxicity, etc. analysis on the whole vaccine construct. [By the way, the authors themselves state that (lines 459-463)].

Response: Thank you for the valid point. The mentioned analyses are done in for the designed vaccine and added to section 3.10 of the revised manuscript.

  1. Report the interactions in the docked complexes (with each of the three receptors considered) in terms of number of hydrogen bonds, salt bridges, etc. A figure illustrating the contact sites should be included.

Response: Thank you for the valid point. However, the authors feel that the provided information is enough as the number of figures and tables are high. The required data is already presented in the manuscript in section 3.12 and 3.13, so we feel any additional data will make the manuscript more complicated.

iii. Double-check the 3D-structure of the cholera toxin b subunit obtained by their modeling against its deposited X-ray diffraction structure.

Response: Yes, confirmed and found Fine. Please note that the 3D structure generated is based on ab initio principle not comparative modeling.

  1. Show that the construct is more effective in the interaction with the chosen receptors than the adjuvant alone.

Response: Interaction between the receptors and vaccine adjuvant molecule is necessary for proper presentation of vaccine epitopes to immune cells. We have seen that both adjuvant and vaccine interact with the receptors as givein in Fig.11.

The authors may consider the following paper as a possible model for their revision: Omoniyi, A.A., Adebisi, S.S., Musa, S.A. et al. In silico design and analyses of a multi-epitope vaccine against Crimean-Congo hemorrhagic fever virus through reverse vaccinology and immunoinformatics approaches. Sci Rep 12, 8736 (2022). https://doi.org/10.1038/s41598-022-12651-1

Response: Thank you for recommending the manuscript. The manuscript was used during revisions and cited in the revised manuscript.

Line 504: "The designed vaccine has a binding affinity with immune receptors". The statement is too generic and not supported by data. Predicting binding affinity from structural models, for instance in terms of dissociation constant, is hampered by the complexity of the interactions, particularly for flexible complexes. The statement should be removed if no quantification can be provided.

Response: The statement is removed as per reviewer suggestion.

OTHER POINTS, to be addressed after the above questions have been fully clarified:

Introduction should be rewritten: unnecessary information should be deleted (e.g., from lines 56-86: for examples see below "minor points"). A line on why Plesiomonas shigelloides is considered an AR bacterial species and a brief discussion on the state-of-the-art of vaccines against Plesiomonas shigelloides, and Shigella, should instead be included (by the way, line 439 needs a reference, as well). The following papers should be considered:

- Michael Janda J., Abbott S.L., McIver C.J., Plesiomonas shigelloides revisited, (2016) Clinical Microbiology Reviews, 29 (2), pp. 349 - 374 DOI: 10.1128/CMR.00103-15

- Selim, S.; Almuhayawi, M.S.; Zakai, S.A.; Salama, A.A.; Warrad, M. Distinction between Antimicrobial Resistance and Putative Virulence Genes Characterization in Plesiomonas shigelloides Isolated from Different Sources. Antibiotics 2022, 11, 85. https://doi.org/10.3390/antibiotics11010085

- Taylor DN, Trofa AC, Sadoff J, Chu C, Bryla D, Shiloach J, Cohen D, Ashkenazi S, Lerman Y, Egan W, et al. Synthesis, characterization, and clinical evaluation of conjugate vaccines composed of the O-specific polysaccharides of Shigella dysenteriae type 1, Shigella flexneri type 2a, and Shigella sonnei (Plesiomonas shigelloides) bound to bacterial toxoids. Infect Immun. 1993 Sep;61(9):3678-87. doi: 10.1128/iai.61.9.3678-3687.1993

- Louis-Antoine Barel & Laurence A. Mulard (2019) Classical and novel strategies to develop a Shigella glycoconjugate vaccine: from concept to efficacy in human, Human Vaccines & Immunotherapeutics, 15:6, 1338-1356, DOI: 10.1080/21645515.2019.1606972

- Herrera, C.M.; Schmitt, J.S.; Chowdhry, E.I.; Riddle, M.S. From Kiyoshi Shiga to Present-Day Shigella Vaccines: A Historical Narrative Review. Vaccines 2022, 10, 645. https://doi.org/10.3390/vaccines10050645

Response: The mentioned references are used in the revised manuscript.

Other approaches should also be briefly mentioned, such as:

Chunjun Qin, Benjamin Schumann, Xiaopeng Zou, Claney L. Pereira, Guangzong Tian, Jing Hu, Peter H. Seeberger, and Jian Yin, Total Synthesis of a Densely Functionalized Plesiomonas shigelloides Serotype 51 Aminoglycoside Trisaccharide Antigen, Journal of the American Chemical Society 2018 140 (8), 3120-3127, DOI: 10.1021/jacs.8b00148

Response: Done in the revised manuscript.

Line 104: please, replace "ready-to-use" with "designed".

Response: Done in the revised manuscript.

Par 2.4 Cd-hit analysis. The statement "sequence identify (identity) of < 30 % and bit score of > 100" should be briefly explained to non-specialists. Actually, this par. should be removed and merged with lines 146-148 of par. 2.6. By the way, on the identification of the virulence factors of P. shigelloides (with reference to the statement of lines 251-253) see: Ekundayo TC, Okoh AI. Pathogenomics of Virulence Traits of Plesiomonas shigelloides That Were Deemed Inconclusive by Traditional Experimental Approaches. Front Microbiol. 2018 Dec 21;9:3077. doi: 10.3389/fmicb.2018.03077.

Response: Added to section 3.14 of the revised manuscript.

Figure 2, 3 and 4 are not useful and should be either removed or moved to the Supporting information. Paragraphs 3.1, 3.2, and 3.3 can be merged into one.

Response: Done in the revised manuscript.

Lines 262-263: "Those proteins were selected showing molecular properties less than 110 KD and instability index less than 40" The readers might not be specialists of the ExPASy server parameters. Significance of the mentioned values must be explained.

Response: Done in the revised manuscript.

Table 2. The values reported for antigenicity (by the way, those from allergenicity and adhesion probability analysis perhaps are missing) should be explained. Lines 274-275 state: "The three proteins were found antigenic, non-allergen and showing adhesion probability value greater than 0.6 indicating good vaccine candidates". I wonder what the values reported in Table 2 refer to? The reasons of choosing that threshold value (0.6) must be explained (the software used for the analysis should be named). The same holds true for Table 3 (how do the reported antigenicity values prove that the selected peptides are good antigens? The reported numbers are not enough) and for the "score 0.8" reported in Par. 3.8, line 283. Please, explain why 0.8.

Response: The mentioned issues are fixed in the revised manuscript.

Line 340: "In multi-peptide base vaccine constructs, some of the peptide residues are enzyme degradable". Proteolytic degradation affects peptide bonds, not single residues that, if anything, may be recognized, not 'degraded'.

Response: The sentence is rephrased in the revised manuscript.

The statement (line 343): "All the residues which are sensitive to enzymatic degradation are replaced with cysteine bond" is meaningless. What do the authors mean for "residues sensitive to enzymatic degradation"? What do the authors mean with "cysteine bond" Could that be that two residues are replaced by a cystine (Cys-S-S-Cys)? Please, use the correct terms.

Response: Corrected in the revised manuscript.

Table 4: The reported values do not help in understanding which residues were actually replaced and why. Sequence number (S.N)? Chi3? Energy? Sum B- factors? The epitope sequences were never numbered in the text. Some residues are apparently making disulfide bonds with two different residues at the same time (what is the difference between "41 Leu" and "41 Leu", third and fourth rows?) The rationale underpinning the chosen substitutions is not clear and should be discussed.

Response: The results are made clear in the revised section of modified manuscript.  

Line 362: "Those solutions were subjected further based on global energy" subjected? Statement not clear.

Response: The sentence is rephrased in the revised manuscript.

Line 420: "A proper increase in the production of adaptive immune response". The statement is too generic.

Response: The sentence is rephrased in the revised manuscript.

Lines 425-427: "Similarly, production of several types of interferon-gamma was also observed which greater than 400000 is for almost 35 days was as mentioned in Figure 13.B." ? Statement not clear.

Response: The sentence is rephrased in the revised manuscript.

Discussion: this session should be more concise.

Response: The discussion section trimmed as per reviewer suggestion.

MINOR POINTS:

Abstract, line 37: "In total of 24 virulent proteins, 03 proteins; flagellar hook.." (?). Statement not clear. Line 40: "[...] and solubility analysis." Please, remove "analysis". line 47, the abbreviation MHC (major histocompatibility complex) should be explained when first encountered in the text.

Response: Done in the revised manuscript.

Introduction, lines 62-71: "About ... respectively". The statement needs a reference. Lines 65-71: "For combating bacterial diseases [...] Boosting host immunity [...] For combating bacterial diseases [...] boost up a human immune response". The same concept - which is by itself not useful to the manuscript - is repeated multiple times. "At present, there is no licensed vaccines are present". Please, correct. Line 74-75: "A vaccine is a biological preparation of weakened, killed, or fragmented microbes that provide active acquired immunity against infectious diseases" This is not the accepted definition of vaccine. Please correct with something like: ": A vaccine is a preparation used to stimulate the body's immune response against diseases. It can be made up of weakened, killed, or fragmented microbes." or, better, remove the statement. Lines 76, 77 and 79: "A study reported by WHO claims that vaccines can prevent 2.5 million deaths per day worldwide."; "A recent epidemiological study..."; "Vaccine saves five lives per minute worldwide." All of those statements should be removed since they do not ground the present study. If maintained, they need supporting reference(s).

Response: The mentioned text is removed/modified in the revised manuscript as per reviewer 1 and reviewer 3 comments.

Table 1. Why the GC content values of the two genomes differ so much in the significant figures?

Lines 285-299 should be moved to the Experimental session.

Response: Moved to supplementary section as per reviewer 3 comment.

Line 303. Please, remove "analysis".

Response: Done in the revised manuscript.

Line 341: "To avoid this problem, disulfide engineering of vaccine construct was for disulfide engineering to supplement the degradable prone regions with disulfide bonds". Please rephrase this statement.

Response: The sentence is rephrased in the revised manuscript.

Lines 442-444: "P. shigelloides in rare cause’s neonatal sepsis and in hospital settings which shows resistance to antibiotics with high mortality and morbidity rate" ? Statement not clear.

Response: The sentence is rephrased in the revised manuscript.

SOME OF THE TYPOS: Line 39 "receptor protein" (not proteins); Line 90 "it screens"; line 97 "mainly causes"; line 122 "that shows sequence..."; Line 384 "to interact" (remove "be"); Line 438: "shows"; Line 454: "is more effective as it screens" (by the way, remove "in specific"). Several more are present in the manuscript.

Response: The manuscript is revised for such typos and corrected.

Reviewer 2 Report

The paper is essentially descriptive and it is not supported either by data or by models.  A good part of the contents refers to known concepts, however the merit consists in having designed a rationale to organize these concepts. Still, the paper needs additional work.

Author Response

Response to Reviewer Comments

We thank the Referee for spending time and interest in our work and for helpful comments that will greatly improve the manuscript. We have checked all the general and specific comments provided by the Referee and have made all the necessary changes according to his indications. Please refer to yellow highlighted sections in the revised manuscript.

Reviewer # 2

Comments and Suggestions for Authors

The paper is essentially descriptive and it is not supported either by data or by models.  A good part of the contents refers to known concepts, however the merit consists in having designed a rationale to organize these concepts. Still, the paper needs additional work.

Response: We understand the reviewer concern. As the scope of the paper is computational, the main objective of the work is to provide theoretical vaccine model for experimentalists to check the designed vaccine immune protective efficacy against the said pathogen in vivo. Reverse Vaccinology (RV), has received more attention in recent years and has been used for the identification of vaccine proteins against different pathogens [1]. The RV approach was first applied to the bacterial pathogen Meningococcus B (MenB) and led to the license Bexsero vaccine [2], where RV played a significant role in screening for an antigen with the broadest bactericidal activity and ultimately resolved the long journey of MenB vaccine development. RV has also been applied to many other bacterial pathogens, including group A Streptococcus, antibiotic-resistant Staphylococcus aureus, Streptococcus pneumonia, and Chlamydia. The efficacy of peptide or subunit based vaccines initially identified through a RV protocol has also been proven experimentall [3,4]. In this study, a RV approach was used to screen possible vaccine proteins against the Plesiomonas shigelloides, identifying Two extracellular proteins, as a strong candidate for vaccine development. Experimental follow up by testing the immune protection efficacy of the screened epitopes in animal models will open for experimentalists and this study will definitely speed up vaccine development process against this pathogen. This text has been highlighted in the introduction section of the revised manuscript.

References

  1. Ong E, Wong MU, Huffman A, He Y. COVID-19 coronavirus vaccine design using reverse vaccinology and machine learning. bioRxiv [Preprint]. 2020 Mar 21:2020.03.20.000141. doi: 10.1101/2020.03.20.000141. Update in: Front Immunol. 2020 Jul 03;11:1581. PMID: 32511333; PMCID: PMC7239068.
  2. Folaranmi T, Rubin L, Martin SW, Patel M, MacNeil JR; Centers for Disease Control (CDC). Use of Serogroup B Meningococcal Vaccines in Persons Aged ≥10 Years at Increased Risk for Serogroup B Meningococcal Disease: Recommendations of the Advisory Committee on Immunization Practices, 2015. MMWR Morb Mortal Wkly Rep. 2015 Jun 12;64(22):608-12. Erratum in: MMWR Morb Mortal Wkly Rep. 2015 Jul 31;64(29):806. PMID: 26068564; PMCID: PMC4584923.
  3. Maione D, Margarit I, Rinaudo CD, Masignani V, Mora M, Scarselli M, Tettelin H, Brettoni C, Iacobini ET, Rosini R, D'Agostino N, Miorin L, Buccato S, Mariani M, Galli G, Nogarotto R, Nardi-Dei V, Vegni F, Fraser C, Mancuso G, Teti G, Madoff LC, Paoletti LC, Rappuoli R, Kasper DL, Telford JL, Grandi G. Identification of a universal Group B streptococcus vaccine by multiple genome screen. Science. 2005 Jul 1;309(5731):148-50. doi: 10.1126/science.1109869. Erratum in: Science. 2013 Jan 11;339(6116):141. Nardi Dei, Vincenzo [corrected to Nardi-Dei, Vincenzo]. PMID: 15994562; PMCID: PMC1351092.
  4. Sette A, Rappuoli R. Reverse vaccinology: developing vaccines in the era of genomics. Immunity. 2010 Oct 29;33(4):530-41. doi: 10.1016/j.immuni.2010.09.017. PMID: 21029963; PMCID: PMC3320742.

Reviewer 3 Report

Introduction

The first two paragraphs are verbose and really out of context and must be deleted.

The objectives of the work must be presented very clearly.

Procedures

In figure 1, please indicate clearly thw correspondence of the various steps with the sub-sections, 2.1., 2.2. etc., of the manuscript.

Results

3.1. Please present the clinical details of the two strains.

3.3. Please add a detailed table in supplementary material in excel form with all the proteins and their details in proteomic analysis.

Did you carry out Gene Ontology for the proteins? If yes, please provide classifications, if no, GO must be performed and the results must be added.

3.8. The MHC allele must be described in supplementary material, not in main text.

Discussion

The discussion must be divided in sub-sections.

The limitations of the study must be transferred as separate sub-section in discussion.

A new sub-section with the clinical significance of the findings must be added.

Author Response

Response to Reviewer Comments

We thank the Referee for spending time and interest in our work and for helpful comments that will greatly improve the manuscript. We have checked all the general and specific comments provided by the Referee and have made all the necessary changes according to his indications. Please refer to yellow highlighted sections in the revised manuscript.

Reviewer # 3

Comments and Suggestions for Authors

Introduction

The first two paragraphs are verbose and really out of context and must be deleted.

Response: The mentioned paragraphs are remodeled in light of reviewer suggestion.

The objectives of the work must be presented very clearly.

 Response: Added to the introduction second paragraph in the revised manuscript.

Procedures

In figure 1, please indicate clearly thw correspondence of the various steps with the sub-sections, 2.1., 2.2. etc., of the manuscript.

Response: Done in the revised manuscript.

Results

3.1. Please present the clinical details of the two strains.

Response: Done in the revised manuscript.

3.3. Please add a detailed table in supplementary material in excel form with all the proteins and their details in proteomic analysis.

Response: Thank you for the comment. It is very difficult to add all proteins analyzed in the study as they are thousands in number. The good vaccine candidates prioritized in the study are listed in Table S.1. 

Did you carry out Gene Ontology for the proteins? If yes, please provide classifications, if no, GO must be performed and the results must be added.

Response: Added to Table 2 of the revised manuscript.

3.8. The MHC allele must be described in supplementary material, not in main text..

Response: Done in the revised manuscript.

Discussion

The discussion must be divided in sub-sections.

Response: Done in the revised manuscript.

The limitations of the study must be transferred as separate sub-section in discussion.

Response: Done in the revised manuscript.

A new sub-section with the clinical significance of the findings must be added.

Response: Done in the revised manuscript.

Round 2

Reviewer 1 Report

The manuscript has been improved. The following statements should be further corrected: 

- Line 96: please, replace "ready-to-use" with "designed".

- line 294: "The antigenic proteins were picked by a value;" This statement must read: "The prediction software assigns an 'antigenicity value' to each antigenic proteins, which correlates with their ability to stimulate immunoresponses"

- lines 296-297 "Proteins with a value > 0.4 are considered to induce strong immunological responses based on published literature." Add a reference to the "published literature". If no published literature for the value 0.4 is available, then please modify the statement to read something like "A value > 0.4 was considered appropriate to identify proteins likely able to induce strong immunological responses". The same reference as lines 296-297 should support lines 268-270 "The antigenic value of greater than 0.4 refers to strong ability of the proteins to induce immune responses compared to those with a value less than 0.4." A reference supporting the threshold value 0.5 for adhesion must also be added (line 271). Again, if no references are available, then the text should read something like: "we consider 0.5 as a reasonable threshold value to identify probably adhesive proteins"

- Line 313: "The designed vaccine was found non-antigenic (0.87)": how comes that a value >0.4 (see above) appoints good candidates and now with 0.87 is considered non-antigenic? The reader might not understand. I hope this is just a typo and "non-" be removed.

- Table 4: I still don't understand if this table means that the software made two disulfide bridges involving Leu41. In the table (rows 3 and 4), there are two pairs or residues: "Thr19-Leu41" and "Ile36-Leu41". How could that be? Perhaps this is easily explained, but I am puzzled.

I still think there are not enough data to support the docking conclusions. Table 5 is really not enough to prove interactions. If anything, at least the authors can define what they consider to be a "strong" interaction? Perhaps the evaluation was based on intermolecular distances? Does the software give an index which arise from a combination of the intermolecular interactions, with a threshold to identify strong interactions? Something must be said to justify the choice of the residues listed in Table 5. The same holds true for Table 6: is it possible that no hydrogen bonds, salt bridges are present, and that Van der Waals interactions are the only favorable contacts?

Author Response

Response to Reviewer Comments

We thank the Referee for spending time and interest in our work and for helpful comments that will greatly improve the manuscript. We have checked all the general and specific comments provided by the Referee and have made all the necessary changes according to his indications. Please refer to green highlighted sections in the revised manuscript.

Reviewer # 1

Comments and Suggestions for Authors

The manuscript has been improved. The following statements should be further corrected: 

- Line 96: please, replace "ready-to-use" with "designed".

Response: Done in the revised manuscript.

- line 294: "The antigenic proteins were picked by a value;" This statement must read: "The prediction software assigns an 'antigenicity value' to each antigenic proteins, which correlates with their ability to stimulate immunoresponses"

Response: Corrected in the revised manuscript.

- lines 296-297 "Proteins with a value > 0.4 are considered to induce strong immunological responses based on published literature." Add a reference to the "published literature". If no published literature for the value 0.4 is available, then please modify the statement to read something like "A value > 0.4 was considered appropriate to identify proteins likely able to induce strong immunological responses". The same reference as lines 296-297 should support lines 268-270 "The antigenic value of greater than 0.4 refers to strong ability of the proteins to induce immune responses compared to those with a value less than 0.4." A reference supporting the threshold value 0.5 for adhesion must also be added (line 271). Again, if no references are available, then the text should read something like: "we consider 0.5 as a reasonable threshold value to identify probably adhesive proteins"

Response: All the revisions are done as per reviewer suggestion.

- Line 313: "The designed vaccine was found non-antigenic (0.87)": how comes that a value >0.4 (see above) appoints good candidates and now with 0.87 is considered non-antigenic? The reader might not understand. I hope this is just a typo and "non-" be removed.

Response: Thank you for highlighting this error and corrected in the revised manuscript.

- Table 4: I still don't understand if this table means that the software made two disulfide bridges involving Leu41. In the table (rows 3 and 4), there are two pairs or residues: "Thr19-Leu41" and "Ile36-Leu41". How could that be? Perhaps this is easily explained, but I am puzzled.

Response: Thank you for highlighting this error. It can be only one and corrected in the revised manuscript.

I still think there are not enough data to support the docking conclusions. Table 5 is really not enough to prove interactions. If anything, at least the authors can define what they consider to be a "strong" interaction? Perhaps the evaluation was based on intermolecular distances? Does the software give an index which arise from a combination of the intermolecular interactions, with a threshold to identify strong interactions? Something must be said to justify the choice of the residues listed in Table 5. The same holds true for Table 6: is it possible that no hydrogen bonds, salt bridges are present, and that Van der Waals interactions are the only favorable contacts?

Response: Thank you for the comment. The shortlisting of the interactions shown in Table 5 is done based on bond distance. Majority of these interactions are within 5 Å.  Similarly, it is clear in Table 6 that van der Waal energy dominate the intermolecular interactions followed by electrostatic energy involving hydrogen bonds. The text is added to the revised manuscript.

Reviewer 2 Report

Ok for the revisions.

Author Response

Response to Reviewer Comments

We thank the Referee for spending time and interest in our work and for helpful comments that will greatly improve the manuscript. We have checked all the general and specific comments provided by the Referee and have made all the necessary changes according to his indications. Please refer to green highlighted sections in the revised manuscript.

Reviewer # 2

Comments and Suggestions for Authors

Ok for the revisions.

Response: Thank you for supporting the revisions and accepting the manuscript for publication.

Reviewer 3 Report

The manuscript has been improved. 
Before acceptance, the authors must add some recent relevant references, hi which are missing. 

Author Response

Response to Reviewer Comments

We thank the Referee for spending time and interest in our work and for helpful comments that will greatly improve the manuscript. We have checked all the general and specific comments provided by the Referee and have made all the necessary changes according to his indications. Please refer to green highlighted sections in the revised manuscript.

Comments and Suggestions for Authors

The manuscript has been improved. 
Before acceptance, the authors must add some recent relevant references, hi which are missing. 

Response: Added to the revised manuscript discussion section.

Round 3

Reviewer 1 Report

The authors addressed my concerns. The paper can be published